# Closed-loop oxygen control for critically ill patients––A systematic review and meta-analysis

**Caroline Gomes Mól** [1☯]*, **Aléxia Gabriela da Silva Vieira** [1☯], **Bianca Maria Schneider Pereira Garcia** [1], **Emanuel dos Santos Pereira** [1], **Raquel Afonso Caserta Eid** [1], **Marcus J. Schultz** [2,3,4,5‡], **Ana Carolina Pereira Nunes Pinto** [6‡], **Ricardo Kenji Nawa** [1☯]*

1 Department of Critical Care Medicine, Hospital Israelita Albert Einstein, São Paulo, SP, Brazil,
2 Department of Intensive Care, Amsterdam UMC, Location AMC, Amsterdam, The Netherlands,
3 Mahidol–Oxford Tropical Medicine Research Unit (MORU), Mahidol University, Bangkok, Thailand,
4 Nuffield Department of Medicine, University of Oxford, Oxford, United Kingdom, 5 Department of Anesthesia, General Intensive Care and Pain Management, Medical University Wien, Vienna, Austria,
6 Iberoamerican Cochrane Center, Biomedical Research Institute Sant Pau, Barcelona, Spain

☯ These authors contributed equally to this work.
‡ These authors also contributed equally to this work.
* caroline.rodrigues@eisntein.br (CGM); ricardo.nawa@einstein.br (RKN)

**Data Availability Statement:** All relevant data are within the manuscript and its Supporting Information files.

## Abstract

### Background

The impact of closed-loop control systems to titrate oxygen flow in critically ill patients, including their effectiveness, efficacy, workload and safety, remains unclear. This systematic review investigated the utilization of closed-loop oxygen systems for critically ill patients in comparison to manual oxygen titration systems focusing on these topics.

### Methods and findings

A search was conducted across several databases including MEDLINE, CENTRAL, EMBASE, LILACS, CINAHL, LOVE, ClinicalTrials.gov, and the World Health Organization on March 3, 2022, with subsequent updates made on June 27, 2023. Evidence databases were searched for randomized clinical parallel or crossover studies investigating closed-loop oxygen control systems for critically ill patients. This systematic review and meta-analysis was performed following the Preferred Reporting Items for Systematic Review and Meta-analysis guidelines. The analysis was conducted using Review Manager software, adopting the mean difference or standardized mean difference with a 95% confidence interval (95% CI) for continuous variables or risk ratio with 95% CI for dichotomous outcomes. The main outcome of interest was the percentage of time spent in the peripheral arterial oxygen saturation target. Secondary outcomes included time for supplemental oxygen weaning, length of stay, mortality, costs, adverse events, and workload of healthcare professional. A total of 37 records from 21 studies were included in this review with a total of 1,577 participants. Compared with manual oxygen titration, closed-loop oxygen control systems increased the percentage of time in the prescribed $SpO_2$ target, mean difference (MD)

**Funding:** The author(s) received no specific funding for this work.

**Competing interests:** MJS was the team leader of Research and New Technologies at Hamilton Medical AG, Bonaduz, Switzerland, from January 2022 until January 2023. The other authors declare no conflicts of interest. This does not alter our adherence to PLOS ONE policies on sharing data and materials. This does not alter our adherence to PLOS ONE policies on sharing data and materials.

25.47; 95% CI 19.7, 30.0], with moderate certainty of evidence. Current evidence also shows that closed-loop oxygen control systems have the potential to reduce the percentage of time with hypoxemia (MD -0.98; 95% CI -1.68, -0.27) and healthcare workload (MD -4.94; 95% CI -7.28, -2.61) with low certainty of evidence.

## Conclusion

Closed-loop oxygen control systems increase the percentage of time in the preferred $SpO_2$ targets and may reduce healthcare workload.

## Trial registration

PROSPERO: CRD42022306033.

## Introduction

Critically ill patients frequently require supplemental oxygen due to the clinical manifestation of inadequate gas exchange [1,2]. Oxygen administration can be considered a lifesaving treatment and may reduce the mortality and morbidity of hypoxemic patients [3,4]. Oxygen is commonly utilized for critically ill patients and its prescription is frequently observed for patients undergoing respiratory support, including mechanical ventilation to correct or prevent hypoxemia [3,5,6].

Hypoxemia can result from many factors, such as lung and cardiovascular diseases, or reduced oxygen levels at high altitudes and can potentially progress to hypoxia, leading to potential organ damage [7–10]. In addition, hyperoxemia, the presence of excess oxygen, also carries risks, including oxidative stress and potential harm to vital organs [11–14]. Thus, finding the right balance is crucial [15–20]. Therefore, healthcare professionals must carefully titrate oxygen to ensure that they are adequate for the patient's needs, minimizing the risks associated with both hypoxemia and hyperoxemia while optimizing patient outcomes.

The literature suggests safe and acceptable targets of peripheral capillary oxygen saturation ($SpO_2$) ranging between 92–98% for patients without lung diseases and 88–92% for patients with previous lung diseases [21]. However, manual adjustment of the fraction of inspired oxygen ($FiO_2$) to precisely deliver oxygen within the target is both challenging and time-consuming for healthcare professionals [22–24]. Therefore, the adoption of automated technologies may be considered to partially reduce workload of healthcare professionals.

Closed-loop oxygen control systems have been developed to provide continuing monitoring and adjustments of oxygen titration based on patients' $SpO_2$, avoiding and treating hypoxemia and hyperoxemia episodes [23,24]. These systems are based on the feedback principle to maintain a target saturation prescribed by continuing titrating oxygen levels [22–24]. It remains uncertain whether closed-loop oxygen control systems are effective, efficient, and safe. It is also uncertain whether these systems affect the workload of healthcare providers. We conducted a systematic review and meta-analysis focusing on these topics.

## Methods

This review is reported in accordance with the Preferred Reporting Items for Systematic Reviews and Meta-Analyses (PRISMA) statement guidelines [25,26]. This systematic review

was registered on PROSPERO [CRD42022306033], and additional details are available in the published protocol [27].

## Search strategy

The search was performed on March 3, 2022, and updated on June 27, 2023, in the MEDLINE, CENTRAL, EMBASE, LILACS, CINAHL, and LOVE evidence databases (**S1–S6 Tables in S1 File**). Furthermore, a search was performed at the ClinicalTrials.gov website and World Health Organization (WHO) International Clinical Trials Registry Platform to find *'ongoing'* and *'unpublished'* studies. There were no restrictions to language, date, or type of publication.

## Eligibility criteria

The eligibility criteria [28] were (P) population: adult ICU patients requiring supplemental oxygen; (I) intervention: any system or device that allows an automatic oxygen titration, including for use with invasive or noninvasive ventilation or with low- or high-flow oxygen therapy; (C) comparator: manual adjustments of oxygen; and (O) outcome: percentage of time in the $SpO_2$ target, time for weaning from oxygen, length of stay, mortality, costs, adverse events, and workload of healthcare professionals. Different records identified by the same registration number were grouped and considered as a single study. This way, we guarantee that studies with multiple publications were included just once.

## Study selection

Two investigators (C.G.M. and A.G.V.) independently screened the studies retrieved by the searches. A third investigator (R.K.N.) was consulted to resolve potential disagreements, if necessary.

## Outcome measures

The primary outcome was the time in the $SpO_2$ target, defined as the percentage of time in which the $SpO_2$ values remained in the predefined range (e.g., 92 to 96% or 88 to 92%). Secondary outcomes included time for weaning from supplemental oxygen, defined as the total time spent in oxygen support during hospitalization, length of stay, mortality, costs, adverse events, and workload of healthcare professionals, defined as the need for the professional to provide oxygen adjustments (e.g., number of manual adjustments; time spent providing adjustments).

## Assessment of characteristics of studies

Characteristics and outcome data from the included studies were independently extracted by two investigators (C.G.M. and A.G.S.) and revised by a third investigator (A.C.P.) using a predefined data collection form.

## Risk of bias

The risk of bias for the outcomes was determined by using the 'Cochrane Risk of Bias 2' (RoB2) tool for randomized and crossover trials [29,30]. The risk of bias arising from the randomization process, deviations from intended interventions, missing outcome data, measurement of the outcome and selection of the reported result were assessed. For crossover studies the risk of bias arising from period and carry-over effects was also assessed.

## Data synthesis and analysis

The mean difference or standardized mean difference was adopted to analyze continuous variables, with a 95% confidence interval [95% CI]. Dichotomous outcomes are presented as risk ratios (RRs) with 95% CIs. When possible, skewed data were adjusted for mean and standard deviation using Wan's methods and the RevMan Calculator [31]. For figures with good resolution, we extracted data using the '*Webplotdigitizer*' website. For crossover studies, the first period was used to perform the analysis to avoid carry-over effects. When substantial heterogeneity was identified ($I^2 \geq 50\%$), we conducted a predefined subgroup analysis for the type of devices. All analyses were performed using Review Manager software (RevMan, Version 5.4.1. Copenhagen: The Nordic Cochrane Centre, The Cochrane Collaboration, 2020).

## Assessment of certainty of evidence

The Grading of Recommendations Assessment, Development and Evaluation (GRADE) system was used to measure and summarize the overall certainty of the current evidence of each outcome [32] through the *'GRADEpro Guideline Development Tool'* software [33].

## Post hoc analysis

Two additional outcomes were added after peer review of the published protocol [27] to provide additional information to analyze the data along with the percentage of time in the $SpO_2$ target, as follows: (a) the percentage of time with hyperoxemia and (b) the percentage of time with hypoxemia. The criteria for hypoxemia and hyperoxemia were considered based on the definitions provided in the studies. We were concerned with carefully verifying the direction of effect and the impact of these data in meta-analyses, describing the contribution of these trials when necessary. In addition, we conducted two posthoc analyses, one comparing the percentage of time in the SpO2 target for subgroups according to the duration of the intervention, and one in subgroups according to the reason for admission, i.e., medical or surgical.

## Results

### Studies included

A total of 14,256 records were identified, and 37 records from 21 studies [34–54] enrolling 1,577 participants were included (**Fig 1** and **Table 1** and **S7–S9 Tables in S1 File**). Of those, 13 studies [34–38,40,42,45,51,52] investigated closed-loop oxygen control systems for use with invasive mechanical ventilation, and 8 studies [41,43,44,46–50] investigated closed-loop oxygen control systems for use with noninvasive respiratory support. Nine studies [36,37,40,44,46–49,54] could be used in the meta-analysis. Further details can be found in the **S10** and **S11 Tables in S1 File**.

### Risk of bias for randomized clinical trials

Most of the outcomes presented some concerns [40,42,46,47,50,51,54] mainly due to insufficient information on the randomization process such as allocation concealment and baseline characteristics (**Fig 2**). Only for costs [48], all domains presented a low risk of bias due to the completed and desirable information provided. For adverse events, length of stay in the intensive care unit (ICU), mortality and time to oxygen weaning, in addition to concerns due to the lack of information from the randomization process, the lack of details on the pre-specified analysis defined a priori motivated the judgment of some concerns among the evaluators [47,50,51,54]. Of note, for adverse events from the seven studies that evaluated this outcome,

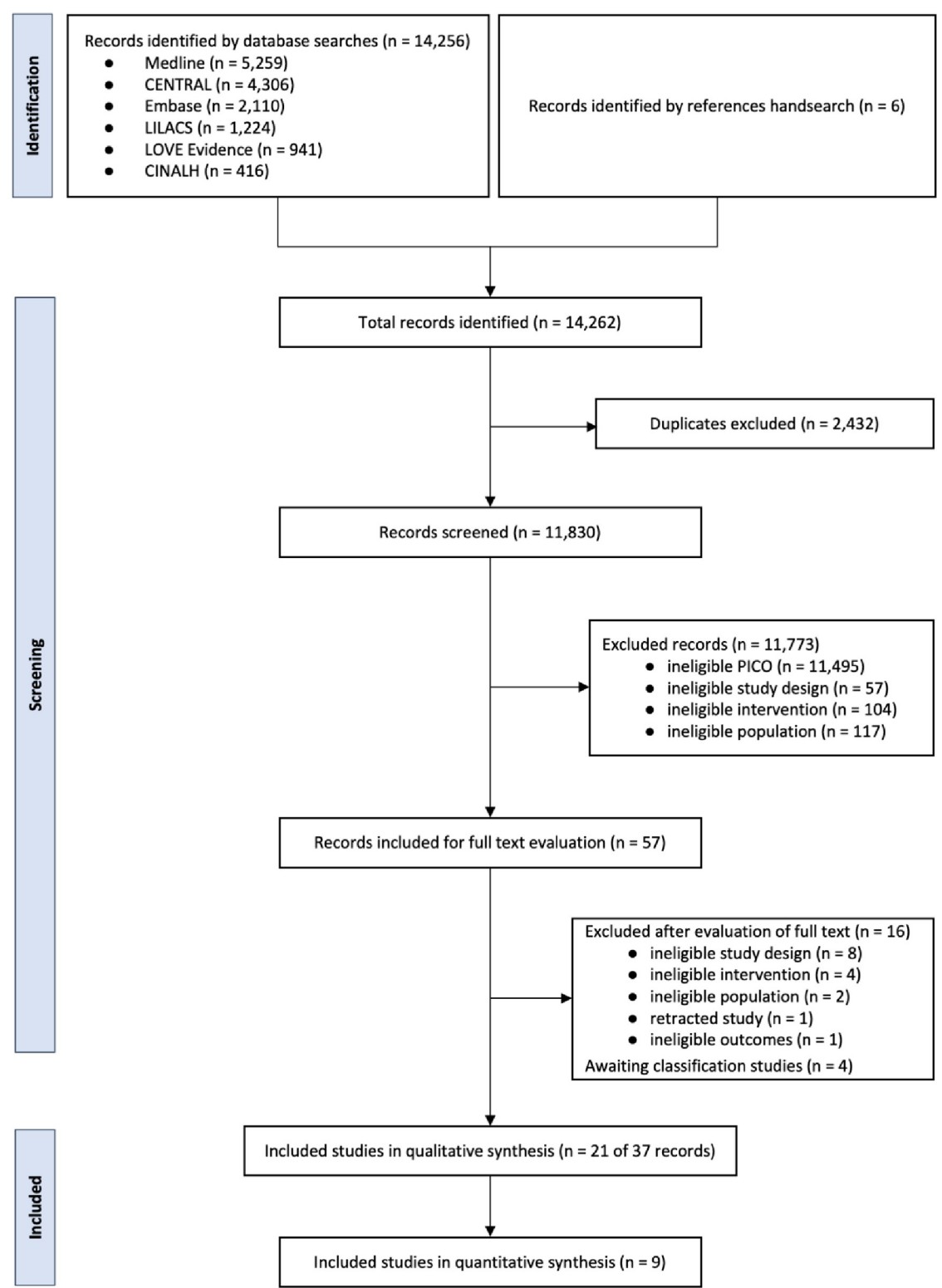

**Fig 1. Flow of trials through the review.** *Abbreviations*: CENTRAL = Cochrane Central Register of Controlled Trials; CINAHL = Cumulative Index to Nursing and Allied Health Literature; EMBASE = Excerpta Medica dataBASE; LILACS = Latin American the Caribbean Literature in Health Sciences; MEDLINE = Medical Literature Analysis and the Retrieval System Online; PICO = Patient Intervention Comparator Outcome.

**Table 1. Summary of included studies.**

| Author (year) Country | Study type | Participants | Intervention | |
|---|---|---|---|---|
| | | | **Exp** | **Con** |
| Arnal et al.[34] (2010) France | RCT crossover | • n = 43<br>• Age, mean (SD), *y* = 64 (15)<br>• Gender = NR<br>• Eligibility = NR<br>• Exclusion criteria = NR | • IMV with closed-loop control of oxygen (INTELLiVENT®–ASV)<br>• Duration = 2-hours<br>• Follow-up = NR | • IMV with manual control of oxygen (ASV mode)<br>• Duration = 2-hours<br>• Follow-up = NR |
| Arnal et al.[35] (2012) France | RCT crossover | • n = 50<br>• Age, mean (SD), *y* = 65 (16)<br>• Gender, *n* (%) = M: 32 (64%) F: 18 (36%)<br>• Eligibility = invasively and passively ventilated patients with moderate severity of ARF.<br>• Exclusion criteria = increased intracranial pressure, severe ARDS requiring a permissive hypercapnia strategy, chronic respiratory failure receiving long-term oxygen therapy and/or home non-invasive ventilation, severe cardiac arrhythmia, therapeutic hypothermia, brain-dead patients, pregnancy and bronchopleural fistula. | • IMV with closed-loop control of oxygen (INTELLiVENT®–ASV)<br>• Duration = 2-hours<br>• Follow-up = NR | • IMV with manual control of oxygen (ASV mode)<br>• Duration = 2-hours<br>• Follow-up = NR |
| Arnal et al.[36] (2018) France | RCT | • n = 60<br>• Age, mean (SD), *y* = Exp 67 (12); Con 62 (16)<br>• Gender, *n* (%) = Exp M: 20 (66.7%), F: 10 (33.3%); Con M: 22 (73.3%), F: 8 (26.7%)<br>• Eligibility = MV patients (expected duration ≥ 48 h)<br>• Exclusion criteria = brain injured patients with GCS < 6, broncho-pleural fistula, chronic or acute dyshemoglobinemia, ventilation drive disorder such as Cheynes Stokes breathing, moribund patient, patient with "Do Not Resuscitate" order before inclusion, chronic respiratory disease requiring long term home ventilation, patient ventilated for planned surgery, pregnancy, lack of informed consent, and the participation in another clinical study. | • IMV with closed-loop control of oxygen (INTELLiVENT®–ASV)<br>• Duration = 4-days<br>• Follow-up = 4-days | • IMV with manual control of oxygen<br>• Duration = 4-days<br>• Follow-up = 4-days |
| Bialais et al.[37] (2016) Belgium | RCT | • n = 70<br>• Age, median [IQR], *y* = Exp 59 [47–69], Con 58 [44–71]<br>• Gender, n = Exp: M: 25; F: 17; Con M: 22; F: 16<br>• Eligibility = MV patients (duration ≥ 48 h) and BMI < 40 kg/m².<br>• Exclusion criteria = ventricular assistance with intra-aortic balloon counterpulsation, bronchopleural fistula, $CO_2$ gradient higher than 15 mmHg, pregnancy, adults under guardianship, people deprived of freedom and inclusion in another study protocol. | • IMV with closed-loop control of oxygen (INTELLiVENT®–ASV)<br>• Duration = 48-hours<br>• Follow-up = NR | • IMV with manual control of oxygen<br>• Duration = 48-hours<br>• Follow-up = NR |
| Buiteman-Kruizinga et al. [53] (2022) Netherlands | RCT crossover | • n = 18<br>• Age, *y* = NR<br>• Gender = NR<br>• Eligibility = patients within 24-hour of start of ventilation and expected to need MV > 24 hours.<br>• Exclusion criteria = MV through a tracheostomy cannula, BMI > 40, any contraindication for use of the automated ventilation mode. | • IMV with closed-loop control of oxygen (INTELLiVENT–ASV)<br>• Duration = 3-hours<br>• Follow-up = NR | • IMV with manual control of oxygen<br>• Duration = 3-hours<br>• Follow-up = NR |

*(Continued)*

**Table 1.** (Continued)

| Author (year) Country | Study type | Participants | Intervention | |
|---|---|---|---|---|
| | | | **Exp** | **Con** |
| Chelly et al.[38] (2020) France | RCT crossover | • n = 265<br>• Age, mean (SD), $y$ = 64 (14)<br>• Gender, $n$ (%) = M: 172 (65); F: 93 (35)<br>• Eligibility = MV for at least 48 h, with a $FiO_2 \leq$ 60%.<br>• Exclusion criteria = prone positioning, use of neuromuscular blocking agents, pregnant women, contraindication to automated ventilation mode (delirium, broncho-pleural fistula, respiratory drive disorder such as Cheyne-Stokes breathing), and low-quality measurement for $SpO_2$. | • IMV with closed-loop control of oxygen (INTELLiVENT®–ASV)<br>• Duration = 6-hours<br>• Follow-up = NR | • IMV with manual control of oxygen<br>• Duration = 6-hours<br>• Follow-up = NR |
| Clavieras et al. [39] (2013) France | RCT crossover | • n = 7<br>• Age, median [IQR], $y$ = 58 [50–64]<br>• Gender, $n$ = M: 10; F: 4<br>• Eligibility = MV patients (spontaneous mode), with an expected duration of MV > 48 h.<br>• Exclusion criteria = clinical instability, decision to withhold life-sustaining treatment, pregnant women. | • IMV with closed-loop control of oxygen (INTELLiVENT®–ASV)<br>• Duration = 24-hours<br>• Follow-up = NR | • IMV with manual control of oxygen<br>• Duration = 24-hours<br>• Follow-up = NR |
| De Bie et al.[40] (2020) Netherlands | RCT | • n = 220<br>• Age, median [IQR], $y$ = Exp 70 [62–76], Con 70 [63–76]<br>• Gender, $n$ (%) = Exp M: 73 (67), F: 36 (33); Con M: 80 (72.1), F: 31 (27.9)<br>• Eligibility = elective cardiac surgery requiring IMV in the ICU.<br>• Exclusion criteria = before surgery: BMI >35 kg m², history of pneumonectomy or lobectomy, COPD (Global Initiative for Chronic Obstructive Lung Disease Class III or IV), already enrolled in another interventional trial. After cardiac surgery: ECMO, hemodynamic instability, fast-track cardiac surgery. | • IMV with closed-loop control of oxygen (INTELLiVENT®–ASV)<br>• Duration = 3-hours<br>• Follow-up = 30-day | • IMV with manual control of oxygen<br>• Duration = 3-hours<br>• Follow-up = 30-day |
| Denault et al. [41] (2020) Canada | RCT crossover | • n = 30<br>• Age, mean (SD), $y$ = 62 (9)<br>• Gender, $n$ (%) = M: 23 (76.7), F: 7 (23.3)<br>• Eligibility = patients with mild and moderate obesity who underwent CABG as sole procedure.<br>• Exclusion criteria = COPD, restrictive syndrome other than one associated with obesity, or obstructive sleep apnea requiring CPAP to focus on obesity as a sole potential risk factor for hyperoxia-induced hypercapnia. | • NIV closed-loop control of oxygen (FreeO2 system) with a $SpO_2$ target of 90 ±2% (conservative)<br>• Duration = 30-min<br>• Follow-up = NR | • Noninvasive manual control of oxygen with a $SpO_2$ target of 95% (liberal)<br>• Duration = 30-min<br>• Follow-up = NR |
| Eremenko et al. [42] (2021) Russia | RCT | • n = 80<br>• Age, mean (SD), $y$ = Exp 59 (8), Con 59.2 (10.8)<br>• Gender, $n$ (%) = M: 57 (71.2), F: 23 (28.8)<br>• Eligibility = aged 30–76 years; uncomplicated cardiac surgery; BMI: 18–35 kg/m²<br>• Exclusion criteria = before surgery: severe renal, hepatic or cardiac failure. After surgery: acute myocardial infarction, hemodynamic instability, $PaO_2/FiO_2 < 150$, allergic reaction, seizures, delirium, acute stroke. | • IMV with closed-loop control of oxygen (INTELLiVENT®–ASV)<br>• Duration = total duration of MV support<br>• Follow-up = hospital length of stay | • IMV with manual control of oxygen<br>• Duration = total duration of MV support<br>• Follow-up = hospital length of stay |

*(Continued)*

**Table 1.** (Continued)

| Author (year) Country | Study type | Participants | Intervention | |
|---|---|---|---|---|
| | | | **Exp** | **Con** |
| Hansen et al.[43] (2018) Denmark | RCT crossover | • n = 19<br>• Age, mean (SD), $y$ = 72.4 (8.4)<br>• Gender, $n$ (%) = M: 12 (60), F: 8 (40)<br>• Eligibility = COPD exacerbation patients with an estimated LOS > 48 h and $SpO_2 \leq 88\%$ on room air.<br>• Exclusion criteria = other significant respiratory or cardiac conditions causing hypoxemia, severe ongoing malignancy, high risk for need of mechanical ventilation. | • Nasal cannula with closed-loop control of oxygen (O2matic system)<br>• Duration = 4-hours<br>• Follow-up = NR | • Nasal cannula with manual control of oxygen<br>• Duration = 4-hours<br>• Follow-up = NR |
| Harper et al.[44] (2021) New Zealand | RCT | • n = 20<br>• Age, mean (SD), $y$ = Exp 68.7 (17), Con 69.2 (19)<br>• Gender, $n$ (%) = Exp M: 6 (60) F: 4 (40), Con M: 6 (60) F: 4 (40)<br>• Eligibility = patients with HFNC with a prescribed $SpO_2$ target ranges from (88–92%) or (92–96%).<br>• Exclusion criteria = incapability of brief interruption in oxygen or high-flow therapy, domiciliary use of CPAP or NIV, obstructive sleep apnoea, respiratory infection or colonization with multidrug resistant bacteria, infection with SARS-CoV-2, haemodynamic instability, end of life care, risk of barotrauma, as assessed by the investigator, nasal or facial conditions precluding use of HFNC, intracranial trauma or trans-nasal neurosurgery (within 6 weeks), any condition which limits the feasibility of continuous SpO2 monitoring, pregnancy or breastfeeding, cognitive impairment or impaired consciousness precluding informed consent, implanted electronic medical device, any other condition which, at the investigator's discretion, is believed may present a safety risk or impact the feasibility of the study or the study results. | • HFNC with closed-loop control of oxygen (Airvo3 system)<br>• Duration = 24-hours<br>• Follow-up = NR | • HFNC with manual control of oxygen (Airvo2 system)<br>• Duration = 24-hours<br>• Follow-up = NR |
| Huynh Ky et al. [50] (2017) Canada | RCT | • n = 60<br>• Age, mean (SD), $y$ = 63 (12)<br>• Gender, $n$ = M: 44, F: 16<br>• Eligibility = patients with acute coronary syndrome.<br>• Exclusion criteria = severe COPD diagnosis. | • Exp 1: NIV closed-loop control of oxygen (FreeO$_2$ system) with 92% of $SpO_2$ target<br>• Exp 2: NIV closed-loop control of oxygen (FreeO2 system) with 96% of $SpO_2$ target<br>• Duration = 24-hours<br>• Follow-up = NR | • Noninvasive manual control of oxygen (FreeO$_2$ system)<br>• Duration = 24-hours<br>• Follow-up = NR |
| Johannigman et al.[45] (2009) USA | RCT crossover | • n = 15<br>• Age, mean (SD), $y$ = 35.6 (13.2)<br>• Gender, $n$ (%) = M: 13 (86.7), F: 2 (13.3)<br>• Eligibility = trauma patients requiring IMV; current $FiO_2$ >35%; indwelling arterial line; and ability to monitor SpO$_2$ accurately<br>• Exclusion criteria = NR | • IMV with closed-loop control of oxygen (Impact 754, Eagle)<br>• Duration = 4-hours<br>• Follow-up = 8-hours | • IMV with manual control of oxygen (Impact 754, Eagle)<br>• Duration = 4-hours<br>• Follow-up = 8-hours |
| Kobayashi et al. [51] (2017) Japan | RCT | • n = 48<br>• Age, $y$ = NR<br>• Gender = NR<br>• Eligibility = patients undergoing cardiac surgery.<br>• Exclusion criteria = NR | • IMV with closed-loop control of oxygen (INTELLiVENT®–ASV)<br>• Duration = 6-hours<br>• Follow-up = NR | • IMV with manual control of oxygen (ASV mode)<br>• Duration = 6-hours<br>• Follow-up = NR |
| Komnov et al. [54] (2023) Russian | RCT | • n = 32<br>• Age, $y$ = NR<br>• Gender = NR<br>• Eligibility: Adult patients (age >30 years), elective cardiac surgery, BMI > 35 kg/m$^2$ and postoperative treatment with MV<br>• Exclusion criteria = NR | • IMV with closed-loop control of oxygen (INTELLiVENT®–ASV)<br>• Duration = total duration of MV support<br>• Follow-up = hospital LOS | • IMV with manual control of oxygen<br>• Duration = total duration of MV support<br>• Follow-up = hospital LOS |

*(Continued)*

**Table 1.** (Continued)

| Author (year) Country | Study type | Participants | Intervention | |
|---|---|---|---|---|
| | | | **Exp** | **Con** |
| L'Her et al.[46] (2017) France | RCT | • n = 187<br>• Age, mean (SD), $y$ = Exp 74.6 (13.2), Con 77.7 (12.2)<br>• Gender, $n$ (%) = Exp M: 56 (60.9), F: 37 (39.1); Con M: 48 (51.1), F: 46 (48.9)<br>• Eligibility = emergency department admission, within < 2 h for acute respiratory disorder requiring ≥ 3 L/min of $O_2$ ($SpO_2$ target ≥ 92%).<br>• Exclusion criteria = life-threatening hypoxaemia, clinical signs of ventilatory assistance requirement, emergent surgery or coronary angiography, pregnant or breastfeeding women, patients under administrative protective measures. | • Mask to administer either low or high $O_2$ flow with closed-loop control of oxygen (FreeO2 system)<br>• Duration = 3-hours<br>• Follow-up = hospital length of stay | • Mask to administer either low or high $O_2$ flow with manual control of oxygen<br>• Duration = 3-hours<br>• Follow-up = hospital length of stay |
| L'Her et al.[47] (2021) France | RCT | • n = 198<br>• Age, mean (SD), $y$ = Exp 62.2 (12.6), Con 63.1 (13.5)<br>• Gender, (%) = Exp M: 55.2, F: 44.8; Con M: 55.3, F: 44.7<br>• Eligibility = patients undergoing abdominal or thoracic surgery; ARISCAT ≥ 26; indication of ventilatory support after surgery.<br>• Exclusion criteria = BMI ≥ 35; obstructive sleep apnea and pregnant patients. | • Nasal cannula/prongs (low flow) or open face mask (low or medium flow) with closed-loop control of oxygen (FreeO2 system)<br>• Duration = 3-hours/3-days<br>• Follow-up = 28-days | • Nasal cannula/prongs (low flow) or open face mask (low or medium flow) with manual control of oxygen<br>• Duration = 3-hours/3-days<br>• Follow-up = 28-days |
| Lellouche et al. [52] (2013) Canada | RCT | • n = 60<br>• Age, mean (SD), $y$ = Exp 67.7 (8.8), Con 65.3 (9.7)<br>• Gender, $n$ (%) = Exp M: 23 (76.7), F: 7 (23.3); Con M: 22 (73.3), F: 8 (26.7)<br>• Eligibility = pre inclusion criteria were: (1) elective cardiac surgery; (2) age 18–90 years; (3) BMI < 40 kg/m$^2$; (4) baseline $PaCO_2$ < 50 mmHg; and (5) serum creatinine < 200 μmol/L. After surgery, hemodynamically stable, and urine output > 50 mL/h.<br>• Exclusion criteria = bronchopleural fistula, study ventilator not available. | • IMV with closed-loop control of oxygen (INTELLiVENT®–ASV)<br>• Duration = 4-hours<br>• Follow-up = ICU length of stay | • IMV with manual control of oxygen<br>• Duration = 4-hours<br>• Follow-up = ICU length of stay |
| Lellouche et al. [48] (2016) Canada | RCT | • n = 50<br>• Age, mean (SD), $y$ = Exp 71 (8), Con 73 (8)<br>• Gender, $n$ (%) = Exp M: 15 (60), F: 10 (40); Con M: 12 (48), F: 13 (52)<br>• Eligibility = hospitalized COPD exacerbation patients with oxygen therapy prescription; age ≥ 40 years; and past or current smoking history of at least 10 pack-years.<br>• Exclusion criteria = admitted for > 24 hours, multidrug-resistant bacteria infection, intermittent noninvasive ventilation, cognitive impairment. | • Nasal cannula or nonocclusive mask with closed-loop control of oxygen (FreeO2 system)<br>• Duration = 7-days<br>• Follow up = 180-days | • Nasal cannula or nonocclusive mask with manual control of oxygen<br>• Duration = 7-days<br>• Follow up = 180-days |

(*Continued*)

**Table 1.** (Continued)

| Author (year) Country | Study type | Participants | Intervention | |
|---|---|---|---|---|
| | | | **Exp** | **Con** |
| Roca et al.[49] (2022) pain | RCT crossover | • n = 45<br>• Age, mean (SD), $y$ = 35.6 (13.2)<br>• Gender, $n$ (%) = M: 24 (53.3), F: 21 (46.7)<br>• Eligibility = ICU admission receiving HFNC with flow rate $\geq$ 40 L/min with $FiO_2 \geq$ 30%, and expected to receive HFNC for at least 8 hours after randomization.<br>• Exclusion criteria = pregnant women, indication or high risk for immediate intubation, indication for non-invasive ventilation, haemodynamic instability, severe acidosis, poor $SpO_2$ signal, chronic or acute dyshaemoglobinaemia, tracheostomy, previously enrolled in this trial or enrolled in another interventional study could not participate. | • HFNC with closed-loop control of oxygen<br>• Duration = 4-hours<br>• Follow-up = NR | • HFNC with manual control of oxygen<br>• Duration = 4-hours<br>• Follow-up = NR |

*Abbreviations*: ALI = acute lung injury; ARDS = acute respiratory distress syndrome; ARF = acute respiratory failure; ARISCAT = assess respiratory risk in surgical patients in Catalonia; ASV = adaptive support ventilation; BMI = body mass index; bpm = beats per minute; CABG = coronary artery bypass grafting; $CO_2$ = carbon dioxide; Con = control; COPD = chronic obstructive pulmonary disease; CPAP = continuous positive airway pressure; DNP = daily nursing procedures; $EtCO_2$ = end-tidal carbon dioxide; Exp = experimental; F = female; $FiO_2$ = inspired fraction of oxygen; GCS = Glasgow coma scale; h = hour; HFNC = high flow nasal cannula; ICU = intensive care unit; IMV = invasive mechanical ventilation; kg = kilogram M = male; Max = maximum; mL = milliliter; Min = minimum; MV = mechanical ventilation; NIV = noninvasive ventilation; NHF = nasal high flow; NR = not reported; $O_2$ = oxygen; PBW = predicted body weight; $P_{INSP}$ = inspiratory pressure; P/F ratio = partial pressure of oxygen/inspired fraction of oxygen ratio; $P_{PEAK}$ = peak pressure; $P_{PLAT}$ = plateau pressure; postop = postoperative; RCT = randomized controlled trial; RR = respiratory rate; ROX = respiratory rate oxygenation; SD = standard deviation; $SpO_2$ = peripheral oxygen saturation; Vt = tidal volume.

in addition to the concerns cited about the randomization process, one study [51] lacks information from blinding of outcome assessors.

### Risk of bias for randomized cross-over trials

Adverse event is the only outcome that included five studies and only one of them [43] had an overall 'low risk' of bias (**Fig 3**). We were concerned about the actual impact of the insufficient information available to assess both proper randomization [34,38,39,45,49,53] and appropriate carry-over time in all outcomes [38,39,45,49,35]. As information from the first phase was not available for most of the crossovers, it was not possible to precisely assess whether there was a deviation from the intended interventions [35,38,43]. We did not find any evidence of missing outcome data or differences between groups for 'measure of outcome' for all outcomes planned in this review and thus considered it as low risk of bias. Furthermore, it was not possible to assess whether a selection of the reported result of the outcomes was due to the absence or insufficient reporting information from the registered protocol [35,45].

### Time spent in predefined $SpO_2$ targets, percentage of time with hypoxemia or hyperoxemia, and weaning

The included studies had different $SpO_2$ targets (**S1 Fig in S1 File**). In 7 studies [37,40,44,46–49], closed-loop oxygen titration devices increased the percentage of time in the predefined $SpO_2$ target with substantial heterogeneity, not completely explained by the planned and post hoc subgroup analysis (**Fig 4, S2 and S3 Figs** and **S11 Table in S1 File**). The sensitivity analysis for the percentage of time in the $SpO_2$ target, included only trials with a low risk of bias (**S4 Fig in S1 File**) and showed similar estimates of the intervention and inconsistency. There's no

| Study ID | Experimental | Comparator | Outcome | D1 | D2 | D3 | D4 | D5 | Overall |
|---|---|---|---|---|---|---|---|---|---|
| Bialais (2016) | Intellivent-ASV® | Conventional ventilatory modes | Adverse Events | + | + | + | + | + | + |
| Eremenko (2021) | Intellivent-ASV® | Conventional ventilatory modes | Adverse Events | ! | + | + | + | + | ! |
| Harper (2021) | Airvo 3 System | Manual oxygen administration | Adverse Events | + | + | + | + | + | + |
| Kobayashi (2017) | Intellivent-ASV® | ASV | Adverse Events | ! | + | + | ! | ! | ! |
| Lellouche (2016) | FreeO₂ | Manual oxygen titration | Adverse Events | + | + | + | + | + | + |
| L'Her (2017) | FreeO₂ | Manual oxygen titration | Adverse Events | ! | + | + | + | + | ! |
| L'Her (2021) | FreeO₂ | Manual oxygen titration | Adverse Events | ! | + | + | + | ! | ! |
| Poder (2018) | FreeO₂ | Manual oxygen titration | Cost | + | + | + | + | + | + |
| Arnal (2018) | Intellivent-ASV® | Conventional ventilatory modes | Healthcare workload | + | + | + | + | + | + |
| Bialais (2016) | Intellivent-ASV® | Conventional ventilatory modes | Healthcare workload | + | + | + | + | + | + |
| Eremenko (2021) | Intellivent-ASV® | Conventional ventilatory modes | Healthcare workload | ! | + | + | + | + | ! |
| Harper (2021) | Airvo 3 System | Manual oxygen titration | Healthcare workload | + | + | + | + | + | + |
| Komnov (2023) | Intellivent-ASV® | Conventional ventilatory modes | Healthcare workload | ! | + | + | + | + | ! |
| Lellouche (2013) | Intellivent-ASV® | Protocolized ventilation group | Healthcare workload | + | + | + | + | + | + |
| Harper (2021) | Airvo 3 System | Manual oxygen administration | Hospital length of stay | + | + | + | + | + | + |
| Bialais (2016) | Intellivent-ASV® | Conventional ventilatory modes | Hospital length of stay | + | + | + | + | + | + |
| Arnal (2018) | Intellivent-ASV® | Conventional ventilatory modes | Hospital length of stay | + | + | + | + | + | + |
| Lellouche (2016) | FreeO₂ | Manual oxygen titration | Hospital length of stay | + | + | + | + | + | + |
| L'Her (2017) | FreeO₂ | Manual oxygen titration | Hospital length of stay | ! | + | + | + | + | ! |
| L'Her (2021) | FreeO₂ | Manual oxygen titration | Hospital length of stay | ! | + | + | + | + | ! |
| Eremenko (2021) | Intellivent-ASV® | Conventional ventilatory modes | ICU and Hospital length of stay | ! | + | + | + | + | ! |
| Arnal (2018) | Intellivent-ASV® | Conventional ventilatory modes | ICU length of stay | + | + | + | + | + | + |
| Bialais (2016) | Intellivent-ASV® | Conventional ventilatory modes | ICU length of stay | + | + | + | + | + | + |
| De Bie (2020) | Intellivent-ASV® | Conventional ventilatory modes | ICU length of stay | ! | + | + | + | + | ! |
| Komnov (2023) | Intellivent-ASV® | Conventional ventilatory modes | ICU length of stay | ! | + | + | + | ! | ! |
| L'Her (2017) | FreeO₂ | Manual oxygen titration | ICU length of stay | ! | + | + | + | + | ! |
| Arnal (2018) | Intellivent-ASV® | Conventional ventilatory modes | Mortality | + | + | + | + | + | + |
| Bialais (2016) | Intellivent-ASV® | Conventional ventilatory modes | Mortality | + | + | + | + | + | + |
| De Bie (2020) | Intellivent-ASV® | Conventional ventilatory modes | Mortality | ! | + | + | + | + | ! |
| Eremenko (2021) | Intellivent-ASV® | Conventional ventilatory modes | Mortality | ! | + | + | + | + | ! |
| Huynh-Ky (2017) | FreeO₂ | Manual oxygen titration | Mortality | ! | + | + | + | ! | ! |
| Komnov (2023) | Intellivent-ASV® | Conventional ventilatory modes | Mortality | ! | + | + | + | ! | ! |
| L'Her (2017) | FreeO₂ | Manual oxygen titration | Mortality | ! | + | + | + | + | ! |
| L'Her (2021) | FreeO₂ | Manual oxygen titration | Mortality (28 days) | ! | + | + | + | + | ! |
| Lellouche (2013) | Intellivent-ASV® | Protocolized ventilation group | Not acceptable zone | + | + | + | + | + | + |
| Lellouche (2016) | FreeO₂ | Manual oxygen titration | Time for O₂ weaning | + | + | + | + | + | + |
| L'Her (2017) | FreeO₂ | Manual oxygen titration | Time for O₂ weaning | ! | + | + | + | + | ! |
| L'Her (2021) | FreeO₂ | Manual oxygen titration | Time for O₂ weaning | ! | + | + | + | + | ! |
| De Bie (2020) | Intellivent-ASV® | Conventional ventilatory modes | Time in the optimal zone until extubation | ! | + | + | + | + | ! |
| Bialais (2016) | Intellivent-ASV® | Conventional ventilatory modes | Time in the SpO₂ target | + | + | + | + | + | + |
| Harper (2021) | Airvo 3 System | Manual oxygen titration | Time in the SpO₂ target | + | + | + | + | + | + |
| Lellouche (2016) | FreeO₂ | Manual oxygen titration | Time in the SpO₂ target | + | + | + | + | + | + |
| L'Her (2017) | FreeO₂ | Manual oxygen titration | Time in the SpO₂ target | ! | + | + | + | + | ! |
| L'Her (2021) | FreeO₂ | Manual oxygen titration | Time in the SpO₂ target | ! | + | + | + | + | ! |
| Lellouche (2013) | Intellivent-ASV® | Protocolized ventilation group | Time in the SpO₂ target | + | + | + | + | + | + |

+ Low risk ! Some concerns High risk

**Fig 2. Risk of bias of RCT included studies.** D1 = randomization process; D2 = deviations from the intended interventions; D3 = missing outcome data; D4 = measurement of the outcome; D5 = selection of the reported results. *Abbreviations*: ASV = adaptive support ventilation; ICU = intensive care unit; PSV = pressure support ventilation; O2 = Oxygen; RoB2 = cochrane risk of bias 2.0 tool for randomized clinical trials; SpO2 = peripheral oxygen saturation.

| Study ID | Experimental | Comparator | Outcome | D1 | DS | D2 | D3 | D4 | D5 | Overall |
|---|---|---|---|---|---|---|---|---|---|---|
| Arnal (2012) | Intellivent-ASV® | Conventional ventilatory modes | Adverse Events | + | ! | ! | + | + | ! | ! |
| Chelly (2020) | Intellivent-ASV® | Conventional ventilatory modes | Adverse Events | ! | + | ! | + | + | + | ! |
| Clavieras (2013) | Intellivent-ASV® | Conventional ventilatory modes | Adverse Events | ! | + | + | + | + | + | ! |
| Hansen (2018) | O₂matic | Manual oxygen titration | Adverse Events | + | + | + | + | + | + | + |
| Johannigman (2009) | Autonomous control of FiO₂ | Manual oxygen titration | Adverse Events | ! | ! | + | + | + | ! | ! |
| Clavieras (2013) | Intellivent-ASV® | Conventional ventilatory modes | Healthcare workload | ! | + | + | + | + | + | ! |
| Roca (2022) | HFNC with automatic oxygen control | Manual oxygen titration | Healthcare workload | ! | + | + | + | + | + | ! |
| Arnal (2012) | Intellivent-ASV® | Conventional ventilatory modes | Healthcare workload | + | ! | ! | + | + | ! | ! |
| Clavieras (2013) | Intellivent-ASV® | Conventional ventilatory modes | Mortality | ! | + | + | + | + | + | ! |
| Johannigman (2009) | Autonomous control of FiO₂ | Manual oxygen titration | Mortality | ! | ! | + | + | + | ! | ! |
| Roca (2022) | HFNC with automatic oxygen control | Manual oxygen titration | Oxygen sonsumption | ! | + | + | + | + | + | ! |
| Johannigman (2009) | Autonomous control of FiO₂ | Manual oxygen titration | Oxygen usage | ! | ! | + | + | + | ! | ! |
| Roca (2022) | HFNC with automatic oxygen control | Manual oxygen titration | Time in the SpO₂ target | ! | + | + | + | + | + | ! |
| Denault (2020) | FreeO₂ | Manual oxygen titration | Time in the SpO₂ target | + | ! | + | + | + | + | ! |
| Johannigman (2009) | Autonomous control of FiO₂ | Manual oxygen titration | Time in the SpO₂ target | ! | ! | + | + | + | ! | ! |
| Chelly (2020) | Intellivent-ASV® | Conventional ventilatory modes | Time in the SpO₂ target | ! | + | ! | + | + | + | ! |
| Hansen (2018) | O₂matic | Manual oxygen titration | Time in the SpO₂ target | + | + | ! | + | + | ! | ! |
| Arnal (2010) | Intellivent-ASV® | ASV | Time in the SpO₂ target | ! | + | ! | + | + | + | ! |
| Buiteman-Kruizinga (2022) | Intellivent-ASV® | Conventional ventilatory modes | Time in the SpO₂ target | ! | ! | ! | + | + | ! | ! |

+ Low risk ! Some concerns - High risk

**Fig 3. Risk of bias of crossover included studies.** D1 = randomization process; DS = bias arising from period and carryover effects; D2 = deviations from the intended interventions; D3 = missing outcome data; D4 = measurement of the outcome; D5 = selection of the reported result. *Abbreviations*: ASV = adaptive support ventilation; ICU = intensive care unit; FiO2 = fraction of inspired oxygen; HFNC = high-flow nasal cannula; PSV = pressure support ventilation; SpO2 = peripheral oxygen saturation.

clinically important difference in effect estimates between the meta-analysis encompassing all studies and the one focusing solely on studies with a low risk of bias, any inconsistency is likely to be insignificant [55]. Despite the observed asymmetry in the funnel plot (**S5 Fig in S1 File**), we cannot definitively attribute it to publication bias due to the limited number of studies included in this outcome. In 6 studies [36,37,40,46,48,49], closed-loop oxygen titration reduced the percentage of time with hypoxemia, with substantial unexplained heterogeneity, due to imprecise results of 2 studies [36,49] (**S6 Fig in S1 File**). In 4 studies [36,46,48,49] closed-loop oxygen titration did not reduce the percentage of time with hyperoxemia, with substantial heterogeneity and inconsistency in 2 studies [36,46] (**S7 Fig in S1 File**).

Four studies [38,43,45,53] were not included in the quantitative analyses due to the absence of reported data from the first crossover period. The mean percentage of time spent in the SpO2 target, ranging from 92 to 96%, was higher in the intervention group when compared to the control group, 83 (SD 21) versus 33 (SD 36), respectively [45]. These findings were similar to one study [43] published, in which the estimated effect presented, a mean difference (MD) of 38.5% [95% CI, 27.8 to 49.3]. One study [38] investigated 265 mechanically ventilated patients during daily nursing procedures. The closed-loop group spent a mean percentage of time in the SpO2 target of 48 (SD 37), compared with the control group of 43 (SD 37). One study [53] investigated the duration during which mechanically ventilated patients maintained their SpO2 within the optimal breath zone, considering the SpO2 target and FiO2. They found

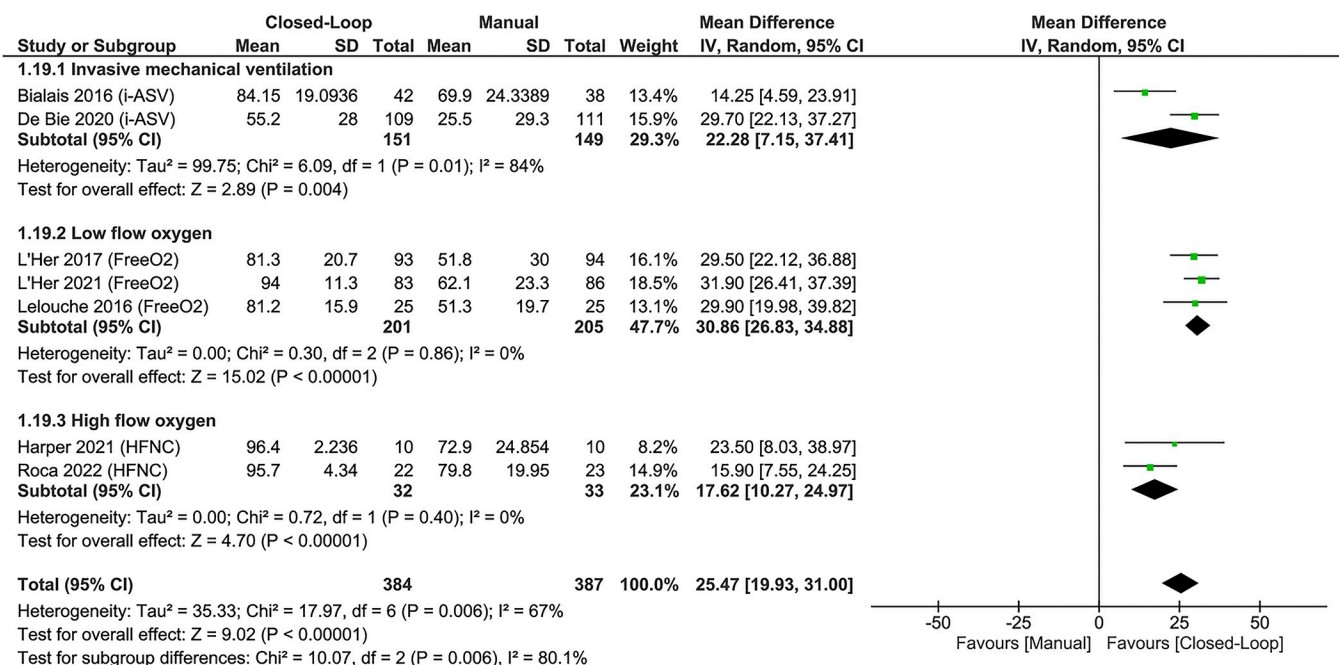

**Fig 4. Subgroup analyses of time spent in the SpO$_2$ target based on pooled data from five studies of noninvasive devices and two studies of mechanical ventilation.** *Abbreviations*: CI = confidence interval; HFNC = high flow nasal cannula; i-ASV = INTELLiVENT-Adaptive Support Ventilation; SD = standard deviation.

that patients in the closed-loop group spent 61% of their time within the optimal oxygenation range, a statistically significant improvement compared to the 52% observed with conventional ventilation. One study [52] evaluated the time spent in the optimal zone (i.e., SpO2 target) in 60 participants. The mean time spent in the closed-loop and control group remained at the optimal zone of 192 (SD 52), and 25 (SD 124) minutes, respectively. This study [52] was not included in the quantitative analysis due to the absence of information on all patients included to change mean and SD data from minutes to percentage of time.

## Clinical outcomes

Of all included studies, 10 studies [36,37,40,42,44,46–49,54] evaluated length of stay, and 10 studies [36,37,39,40,42,45,47,49,50,54] evaluated mortality. In most studies, the intervention was not applied throughout the total duration of oxygen supplementation (**S11 Table in S1 File**). We considered it inappropriate to meta-analyze these details.

## Costs

Of all studies, only one study [48] investigated hospitalization costs, and 2 studies [45,49] evaluated oxygen consumption. These studies suggest that closed-loop oxygen titration systems may result in a reduction in costs and oxygen consumption (**S11 Table in S1 File**).

## Adverse events

Of all studies, 14 studies [34,35,37–39,42–48,51,52] reported adverse events. Quantitative analyses were not performed due to the wide variety and the absence of a standardized definition of adverse events (**S11** and **S12 Tables in S1 File**).

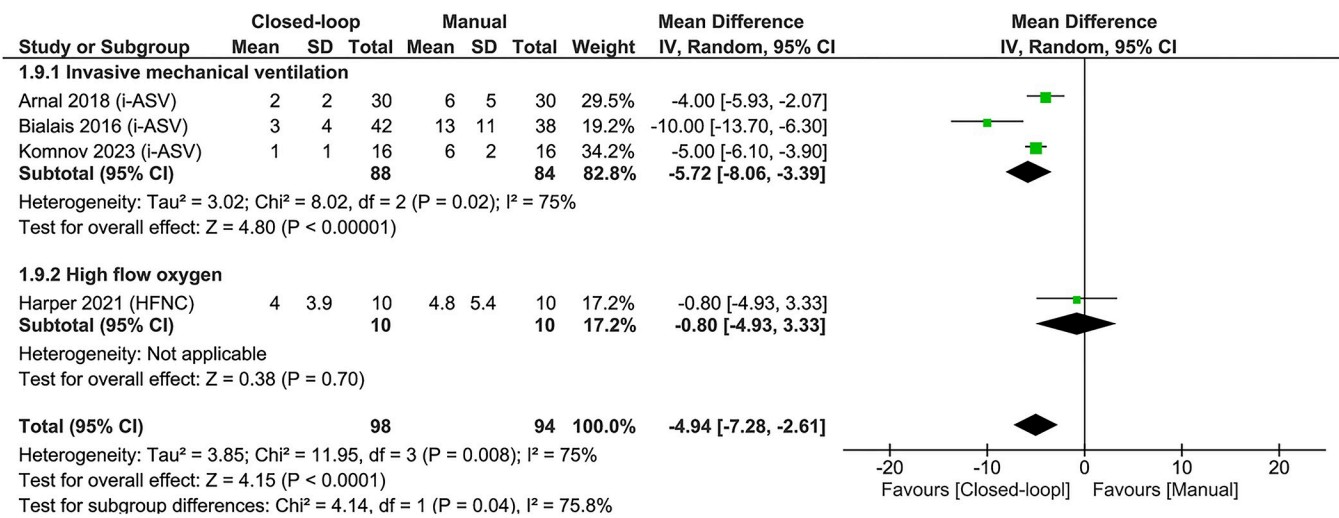

**Fig 5. Forest plot for workload of healthcare professionals.** *Abbreviations*: CI = confidence interval; HFNC = high flow nasal cannula; i-ASV = INTELLiVENT-Adaptive Support Ventilation; SD = standard deviation.

### Healthcare workload

Of the 7 studies [35–37,42,44,49,54] that evaluated the workload of healthcare professionals, 4 studies [36,37,44,54] were included in a quantitative analysis, suggesting that closed-loop oxygen control may reduce healthcare workload (**Fig 5**). In the sensitivity analyses excluding trials with some concerns according to the risk of bias assessment (**S8 Fig in S1 File**), we observed a slight increase in the heterogeneity ($I^2$ = 83%) maintaining the direction of the effect of interventions (MD -4.94, 95% CI -9.43 to -0.46).

### Discussion

The results of this systematic review and meta-analysis show that closed-loop oxygen control devices may increase the percentage of time in the $SpO_2$ target and reduce the percentage of time with hypoxemia. Closed-loop oxygen titration was found not to be associated with a lower percentage of time with hyperoxemia, time for weaning supplemental oxygen, adverse events, length of stay, and mortality. The evidence suggests that closed-loop oxygen titration may reduce costs, oxygen consumption and healthcare workload.

Although both are harmful, episodes of hypoxemia (i.e., $SpO_2$ < 90 to 92%) are frequently regarded as a *'red flag'* when compared with hyperoxemia episodes (i.e., $SpO_2$ > 94–96 to 100%). Consequently, supplemental oxygen is typically considered to be widely available and liberally administered, with little to no attention given to avoiding excessive use. Closed-loop systems for oxygen titration seem to increase the time spent in the $SpO_2$ target, contributing to the correct dose adjustment of oxygen and avoiding hypoxia episodes. However, it may not effectively reduce the occurrence of hyperoxemia, particularly in mechanical ventilation systems, or when high oxygen targets are being used. For instance, one trial [36] showed a high incidence of hyperoxemia with automatic oxygen control; however, this incidence was also high in the control group (35% vs. 33%). The included studies reported the use of different upper limits of $SpO_2$ target and timing of intervention, making it difficult to compare the results. In this scenario, the predefined width range could have influenced the results of the percentage of time in the $SpO_2$ target, hyperoxemia and hypoxemia. Substantial heterogeneity was also found for the percentage of time spent in the $SpO_2$ target, not fully explained by

subgroup analysis. The duration of the intervention and the different patient characteristics, could explain the substantial heterogeneity observed among the studies included in the subgroup analysis of patients undergoing mechanical ventilation support.

Supplemental oxygen use is frequently observed in patients admitted to hospitals and is considered one of the most common treatments provided to critically ill patients. The underrecognition and delay in the correction of hypoxemia remains a major problem in clinical practice associated with an increase in mortality [56], especially in settings where resources are limited and in countries or regions [57]. In addition to socioeconomic factors, the health crisis triggered by the coronavirus disease (COVID-19) pandemic highlighted inequities related to an unprecedented need for oxygen supply and other resources, such as ICU beds, specialized professionals, invasive and noninvasive ventilators, and other medical supplies [58], and consequently, the indices of mortality in patients with and without COVID-19 significantly increased [59,60].

Investments in technology to develop new innovative devices for closed-loop oxygen titration that can efficiently manage resources have grown exponentially in recent years for both clinical and research uses. This was also observed in our findings, as most of the included trials were industry-sponsored investigations. Furthermore, unusual situations, such as the COVID-19 pandemic highlighted the importance and need to rationally use oxygen in clinical practice, as prolonged use of unnecessary supplemental oxygen increases health costs, ICU, and hospital stays, exposing patients to infections and psychosocial complications resulting from the hospitalization process. Despite the benefits of closed-loop devices for oxygen titration, the clinical use at bedside is still limited, due to the high initial investment necessary to acquire such technology. The direct cost related to closed-loop oxygen titration systems was evaluated in only one study [48], and oxygen consumption, an indirect measure of cost, was evaluated in two studies [45,49], showing inaccurate results due to the small number of included patients. Future studies should investigate 'cost outcome' to assist decision-makers in providing an assertive plan for closed-loop oxygen titration device implementation for clinical and research uses.

Patients admitted to the ICU are frequently exposed to different procedures, such as physical therapy and rehabilitation interventions (e.g., airway clearance, suctioning, and positioning), which can promote respiratory and hemodynamic changes according to their clinical condition. In addition, lung dynamics are directly affected by the patient's clinical presentation. Thus, it can require frequent parameter adjustments during invasive or noninvasive ventilation support to maintain the predefined target of $SpO_2$, increasing the workload of healthcare professionals. Closed-loop devices cannot replace healthcare professionals as they cannot handle complex clinical changes in oxygen adjustments. However, they can ease the workload of professionals by automating oxygen adjustments and freeing up time for patient care. These devices do not eliminate the need for clinical discussions and adjustments, especially for significant changes. They require alarms and data visualization for informed decision-making. While their safety is not clearly defined, they offer a potential way to reduce bedside tasks; however, more research on adverse events is needed due to oxygen-related risks. The identification and reporting processes of adverse events can be challenging. Researchers may consider the inclusion of adverse events as an outcome, reporting additional details of the definition, occurrence, and classification of events.

## Strengths and limitations

This review has strengths that have helped reduce potential bias in the review process, such as a thorough literature search, rigorous and well-established methods to minimize bias, including multiple reviewers to independently screen abstracts, review studies, extract data, assess

the risk of bias and the certainty of evidence outlined by the Cochrane Collaboration [30]. In addition, primary study authors were contacted to provide additional information, data, or clarifications when needed. For crossover randomized controlled trials (RCTs), we also contacted authors to request data from the first phase of interventions to describe baseline characteristics and results separately from patients randomized to each stage. We, therefore, believe that it is unlikely that we missed any relevant trials. Thus, this systematic review and meta-analysis provided important insights. Additional studies investigating the use of closed-loop oxygen systems through the entire duration of oxygen support should be developed to improve the findings for certainty of evidence.

However, this review has some limitations. First, due to the short intervention period in most studies it was difficult to establish a direct relationship between the device and the patient's length of stay, mortality and time for oxygen weaning. It is important to emphasize that most studies in this meta-analysis featured brief intervention periods, ranging from only a few hours to 24 hours. This limitation diminishes the generalizability of the results to patients necessitating prolonged oxygen therapy, typically those in the most critical condition. Second, despite our considerable efforts to mitigate publication bias, we were unable to reliably assess it through the funnel plot due to the limited number of available studies. Third, there was insufficient information about the randomization process of some studies, so we recommended the use of reliable methods (e.g., computer-generated random numbers) and detailed reporting to ensure homogeneity between groups. Fourth, the washout period between interventions of crossover RCTs was short and heterogeneous, and the possibility of the carry-over effect cannot be ruled out. Although the devices provide the same substance (oxygen), the type of device can influence the patient's stability. Fifth, the findings of the invasive mechanical ventilation subgroup should be interpreted with caution. Automatic mechanical ventilation modes adjust a group of ventilation settings such as tidal volume, respiratory rate, $FiO_2$, and positive end-expiratory pressure. Thus, the findings of this review regarding the percentage of time spent in the $SpO_2$ target cannot be solely attributed to the oxygen titration. Additionally, the number of adjustments in ventilation parameters may also have influenced healthcare workload outcomes.

In summary, while closed-loop oxygen titration systems probably increase the percentage of time spent within desired $SpO_2$ ranges and reduce healthcare workload and costs, their safety remains uncertain. Adverse events are likely underreported in existing studies. Moreover, the extent to which these systems enhance efficiency, such as by reducing the duration of oxygen weaning, and even more importantly, by affecting length of stay and mortality rates, remains unclear. This uncertainty primarily stems from the limited duration of the intervention in the scarce, and small existing studies. Future research should focus on evaluating effectiveness, safety, and efficacy over longer periods, and prioritize patient-centered endpoints.

## Supporting information

**S1 File. Additional analysis, detailed search strategies, and supplementary information on the included studies.**
(DOCX)

## Acknowledgments

The authors acknowledge Dr Thiago Domingos Corrêa for providing administrative support. We would like to thank Helena Spalic for the valuable contribution proofreading this systematic review.

## Author Contributions

**Conceptualization:** Caroline Gomes Mól, Aléxia Gabriela da Silva Vieira, Ricardo Kenji Nawa.

**Data curation:** Caroline Gomes Mól, Aléxia Gabriela da Silva Vieira, Bianca Maria Schneider Pereira Garcia, Emanuel dos Santos Pereira, Raquel Afonso Caserta Eid, Ricardo Kenji Nawa.

**Formal analysis:** Caroline Gomes Mól, Aléxia Gabriela da Silva Vieira, Marcus J. Schultz, Ana Carolina Pereira Nunes Pinto, Ricardo Kenji Nawa.

**Investigation:** Caroline Gomes Mól, Aléxia Gabriela da Silva Vieira, Marcus J. Schultz, Ana Carolina Pereira Nunes Pinto, Ricardo Kenji Nawa.

**Methodology:** Caroline Gomes Mól, Aléxia Gabriela da Silva Vieira, Bianca Maria Schneider Pereira Garcia, Emanuel dos Santos Pereira, Raquel Afonso Caserta Eid, Marcus J. Schultz, Ana Carolina Pereira Nunes Pinto, Ricardo Kenji Nawa.

**Project administration:** Caroline Gomes Mól, Aléxia Gabriela da Silva Vieira, Ricardo Kenji Nawa.

**Supervision:** Caroline Gomes Mól, Aléxia Gabriela da Silva Vieira, Ricardo Kenji Nawa.

**Validation:** Caroline Gomes Mól, Aléxia Gabriela da Silva Vieira, Bianca Maria Schneider Pereira Garcia, Emanuel dos Santos Pereira, Raquel Afonso Caserta Eid, Marcus J. Schultz, Ana Carolina Pereira Nunes Pinto, Ricardo Kenji Nawa.

**Visualization:** Caroline Gomes Mól, Aléxia Gabriela da Silva Vieira, Ricardo Kenji Nawa.

**Writing – original draft:** Caroline Gomes Mól, Aléxia Gabriela da Silva Vieira, Bianca Maria Schneider Pereira Garcia, Emanuel dos Santos Pereira, Raquel Afonso Caserta Eid, Marcus J. Schultz, Ana Carolina Pereira Nunes Pinto, Ricardo Kenji Nawa.

**Writing – review & editing:** Caroline Gomes Mól, Aléxia Gabriela da Silva Vieira, Marcus J. Schultz, Ana Carolina Pereira Nunes Pinto, Ricardo Kenji Nawa.

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
