## [Decision Letter · Decision Letter 0]

19 Mar 2024

PONE-D-24-04130Closed-loop oxygen control for critically ill patients––A systematic review and meta-analysisPLOS ONE

Dear Dr. Mól,

Thank you for submitting your manuscript to PLOS ONE. After careful consideration, we feel that it has merit but does not fully meet PLOS ONE’s publication criteria as it currently stands. Therefore, we invite you to submit a revised version of the manuscript that addresses the points raised during the review process.

**ACADEMIC EDITOR: **The manuscript addresses an important issue pertinent to management of patients needing oxygen therapy; hence, may be an useful publication. However, the manuscript need major revision. Please see the comments below and also address issues raised by reviewers. The language need thorough check for grammar and syntax.Also, it would be better if main results can be presented as main text, instead of supplementary.Please submit your revised manuscript by May 03 2024 11:59PM. If you will need more time than this to complete your revisions, please reply to this message or contact the journal office at plosone@plos.org. Please include the following items when submitting your revised manuscript:A rebuttal letter that responds to each point raised by the academic editor and reviewer(s). You should upload this letter as a separate file labeled 'Response to Reviewers'.A marked-up copy of your manuscript that highlights changes made to the original version. You should upload this as a separate file labeled 'Revised Manuscript with Track Changes'.An unmarked version of your revised paper without tracked changes. You should upload this as a separate file labeled 'Manuscript'.

We look forward to receiving your revised manuscript.

Kind regards,

Vijay Hadda, MD

Academic Editor

PLOS ONE

“MJS was the team leader of Research and New Technologies at Hamilton Medical AG, Bonaduz, Switzerland, from January 2022 until January 2023. The other authors declare no conflicts of interest.”

Reviewers' comments:

Reviewer's Responses to Questions

**Comments to the Author**

1. Is the manuscript technically sound, and do the data support the conclusions?

Reviewer #1: Yes

Reviewer #2: Yes

Reviewer #3: Partly

2. Has the statistical analysis been performed appropriately and rigorously? 

Reviewer #1: Yes

Reviewer #2: Yes

Reviewer #3: No

3. Have the authors made all data underlying the findings in their manuscript fully available?

Reviewer #1: Yes

Reviewer #2: Yes

Reviewer #3: Yes

4. Is the manuscript presented in an intelligible fashion and written in standard English?

Reviewer #1: Yes

Reviewer #2: Yes

Reviewer #3: No

5. Review Comments to the Author

Reviewer #1: The meta analysis has been thoroughly conducted but there are various issues which need consideration

1 Most studies have a very short duration of intervention ranging for only few hours which will make it difficult to generalise the results for patients requiring oxygen for prolonged duration

2 Exclusion criterion in various studies can be included

3 Hyperoxeia is equally detrimental specially in patients with Type 2 respiratory failure patients. What were the causes of hyperoxia in closed loop systems should be briefly discussed

4 Subgroup analysis based on duration of intervention should be done as there are various studies having intervention for more than 24 hours.

5 Separate analysis for studies with post surgical patients and patients with medical conditions requiring supplemental oxygen should be considered

Reviewer #2: The aim of your study was to evaluate whether closed-loop oxygen control systems are effective, efficient and safe.

Despite heterogeneity of studies it was seen that Closed-loop oxygen titration are effective in reducing costs, oxygen consumption and healthcare workload.

Are they safe - No; as they resulted in longer patient times with hyperoxemia

Are they efficient -No; as time for weaning from supplemental oxygen's not shown to reduce with these systems and they had no effects on ventilator associated adverse events, length of stay, and mortality.

Please include the same clearly in conclusions.

Reviewer #3: The authors have made a substantial effort to collate the literature and come to a reasonable conclusion. However, this meta analysis does not have a funnel plot to look at the publication bias. There is considerable heterogeneity(I2=80.1%) and lacks a sensitivity analysis.

6. PLOS authors have the option to publish the peer review history of their article (what does this mean?). If published, this will include your full peer review and any attached files.

Reviewer #1: No

Reviewer #2: **Yes: **Prof Manu Chopra

Reviewer #3: No

---

## [Author Response · Author response to Decision Letter 0]

19 Apr 2024

Dear prof. Vijay Hadda, dear editor,

We sincerely appreciate the opportunity to submit the revised version of our manuscript entitled ‘Closed-loop Oxygen Control for Critically Ill Patients: A Systematic Review and Meta-analysis’.

Your comments and those of the respected reviewers were very helpful for revising the work, we have meticulously implemented changes to enhance the quality and clarity of our manuscript.

Below, we provide a detailed point-by-point response to the reviewer’s comments.

We look forward to your response, and also the responses from the reviewers if they get a second chance to review our work.

Kindest regards, and on behalf of all coauthors,

Caroline Gomes Mól

 

ACADEMIC EDITOR: 

1. The language need thorough check for grammar and syntax.

Thank you for your comment. We have carefully reviewed both the manuscript and supporting information, conducting a thorough check for grammar and syntax.

2. Also, it would be better if main results can be presented as main text, instead of supplementary.

Thank you for your comment. As suggested, we have transferred certain topics and figures from the supplementary material to the main text, as outlined below:

● [Page 9; line 10] S1 Appendix. Overall risk of bias for each planned outcome.

● [Page 9; line 13; Figure 2; ‘5. [PLoS One - Major Revision] Fig 2 Rob RCT’] S1 Fig. Risk of bias of RCT included studies.

● [Page 9; line 24; Figure 3; ‘6. [PLoS One - Major Revision] Fig 3 Rob RCT crossover’] S2 Fig. Risk of bias of crossover included studies.

● [Page 11; line 3] S3 Appendix. Summary of studies not included in the quantitative analyses.

● [Page 12; line 15; ‘7. [PLoS One - Major Revision] Fig 5’] S10 Fig. Forest plot of healthcare professionals' workload.

Reviewer 1

1. Most studies have a very short duration of intervention ranging for only few hours which will make it difficult to generalize the results for patients requiring oxygen for prolonged duration.

Thank you for your comment. This is a very valid and important comment. We added this limitation to the limitations section of the Discussion of the revised version of the manuscript, as follows:

● [Page 16; line 13] ‘It is important to emphasize that most studies in this meta-analysis featured brief intervention periods, ranging from only a few hours to 24 hours. This limitation diminishes the generalizability of the results to patients necessitating prolonged oxygen therapy, typically those in the most critical condition.’ 

2. Exclusion criterion in various studies can be included.

We appreciate your suggestions. Exclusion criteria of studies are now provided in table 1 as follows:

3. Hyperoxemia is equally detrimental especially in patients with Type 2 respiratory failure patients. What were the causes of hyperoxemia in closed-loop systems should be briefly discussed.

Thank you for your comment and question. The percentage of time with hyperoxemia ranged from less than 1% to 35% in the closed-loop group and less than 1% to 31% in the manual titration group, and this was found to be not statistically different. Of note, in most of the studies, the duration of hyperoxemia was short. In the one study with a high percentage of time with hyperoxemia (Arnal, 2018), utilization of INTELLiVENT–ASV resulted in 35% of time with hyperoxemia versus 31.33% with manual titration. We added the following to the revised version of the manuscript:

● [Page 13; line 15] ‘However, it may not effectively reduce the occurrence of hyperoxemia, particularly in mechanical ventilation systems, or when high oxygen targets are being used. For instance, one trial [36] showed a high incidence of hyperoxemia with automatic oxygen control; however, this incidence was also high in the control group (35% vs. 33%).’

4. Subgroup analysis based on duration of intervention should be done as there are various studies having intervention for more than 24 hours.

Thank you for this valuable suggestion. See also our response to your comment 5, below. According to this suggestion, and also your suggestion in comment 5, we conducted subgroup analyses considering the duration of the intervention, and type of ICU admission, focusing on the primary endpoint of our study. We added the findings as a post hoc analysis to the supplement, and added the following to the revised version of the manuscript:

● [Page 8; line 21] ‘In addition, we conducted two posthoc analyses, one comparing the percentage of time in the SpO2 target for subgroups according to duration of the intervention, and one in subgroups according to the reason for admission, i.e., medical or surgical.’

● [Page 10; line 11] ‘In 7 studies [37,40,44,46–49], closed-loop oxygen titration devices increased the percentage of time in the predefined SpO2 target with substantial heterogeneity, not completely explained by the planned and post hoc subgroup analysis (Fig 4, S2 Fig, S3 Fig and S11 Table).’

● [Supporting information page 5] ‘S2 Fig. Forest plot of subgroup analysis of duration of intervention for the percentage of time spent in the SpO2 target.’

● [Supporting information page 6] ‘S3 Fig. Forest plot of subgroup analysis of patient’s condition (medical or post-surgical) for the percentage of time spent in the SpO2.'

5. Separate analysis for studies with post-surgical patients and patients with medical conditions requiring supplemental oxygen should be considered.

See our reply to your comment 4, above.

Reviewer 2

1. The aim of your study was to evaluate whether closed-loop oxygen control systems are effective, efficient and safe. Despite heterogeneity of studies it was seen that Closed-loop oxygen titration are effective in reducing costs, oxygen consumption and healthcare workload. Are they safe - No; as they resulted in longer patient times with hyperoxemia. Are they efficient - No; as time for weaning from supplemental oxygen's not shown to reduce with these systems and they had no effects on ventilator associated adverse events, length of stay, and mortality. Please include the same clearly in conclusions.

Thank you for your comments and suggestions. We agree that there remains a substantial gap in our understanding of the safety and efficacy of closed-loop oxygen systems in reducing mortality rates and shortening hospital stays.

We revised the conclusion to better address these concerns raised, as follows:

● [Page 17; line 6]‘In summary, while closed-loop oxygen titration systems probably increase the percentage of time spent within desired SpO2 ranges and reduce healthcare workload and costs, their safety remains uncertain. Adverse events are likely underreported in existing studies. Moreover, the extent to which these systems enhance efficiency, such as by reducing the duration of oxygen weaning, and even more importantly, by affecting length of stay and mortality rates, remains unclear. This uncertainty primarily stems from the limited duration of the intervention in the scarce, and small existing studies. Future research should focus on evaluating effectiveness, safety, and efficacy over longer periods, and prioritize patient-centered endpoints.’

Reviewer 3

1. This meta-analysis does not have a funnel plot to look at the publication bias. There is considerable heterogeneity (I2=80.1%) and lacks a sensitivity analysis.

Thank you for your comment regarding the 'funnel plot' and sensitivity analysis. We appreciate and agree with your suggestion. We have incorporated sensitivity analysis for the main outcomes, as follows:

● [Page 10; line 14] ‘The sensitivity analysis for the percentage of time in the SpO2 target, included only trials with a low risk of bias (S4 Fig.) and showed similar estimates of the intervention and inconsistency. There's no clinically important difference in effect estimates between the meta-analysis encompassing all studies and the one focusing solely on studies with a low risk of bias, any inconsistency is likely to be insignificant [55].’ 

[Supporting information page 7] S4 Fig. Sensitivity analyses for the percentage of time spent in the SpO2 target 

[Page 12; line 15] ‘In the sensitivity analyses excluding trials with some concerns according to the risk of bias assessment (S8 Fig), we observed a slight increase in the heterogeneity (I² = 83%) maintaining the direction of the effect of interventions (MD -4.94, 95% CI -9.43 to -0.46).’

[Supporting information page 13] S8 Fig. Sensitivity analyses for healthcare professional’s workload 

We fully agree that the absence of a Funnel Plot analysis is a limitation of our systematic review. Despite the inclusion of only seven studies in the meta-analysis, we also performed the funnel plot for the percentage of time in the SpO2 target (see the figure below). However, the test for publication bias is severely underpowered for this outcome, and visual inspection of a funnel plot with few studies can be highly misleading. Thus, this result should be interpreted with caution. We have incorporated this information into the results and discussion section, as follows:

● [Page 10; line 19] ‘Despite the observed asymmetry in the funnel plot (S5 Fig.), we cannot definitively attribute it to publication bias due to the limited number of studies included in this outcome.’

● [Page 16; line 16] ‘Second, despite our considerable efforts to mitigate publication bias, we were unable to reliably assess it through the funnel plot due to the limited number of available studies.’

[Supporting information page 8] S5. Fig. Funnel plot for the percentage of time spent in the SpO2 target.

Your advice aligns with the suggestions made by other reviewers above, where subgroup analyses have been proposed with the aim of exploring how effect varies across different groups. (please refer to our response to comments 4 and 5 by reviewer '1').

---

## [Decision Letter · Decision Letter 1]

17 May 2024

Closed-loop oxygen control for critically ill patients––A systematic review and meta-analysis

PONE-D-24-04130R1

Dear Dr. Mól,

We’re pleased to inform you that your manuscript has been judged scientifically suitable for publication and will be formally accepted for publication once it meets all outstanding technical requirements.

Kind regards,

Vijay Hadda, MD

Academic Editor

PLOS ONE

Additional Editor Comments (optional):

Reviewers' comments:

Reviewer's Responses to Questions

**Comments to the Author**

1. If the authors have adequately addressed your comments raised in a previous round of review and you feel that this manuscript is now acceptable for publication, you may indicate that here to bypass the “Comments to the Author” section, enter your conflict of interest statement in the “Confidential to Editor” section, and submit your "Accept" recommendation.

Reviewer #1: All comments have been addressed

Reviewer #2: All comments have been addressed

2. Is the manuscript technically sound, and do the data support the conclusions?

Reviewer #1: Yes

Reviewer #2: Yes

3. Has the statistical analysis been performed appropriately and rigorously? 

Reviewer #1: Yes

Reviewer #2: Yes

4. Have the authors made all data underlying the findings in their manuscript fully available?

Reviewer #1: Yes

Reviewer #2: (No Response)

5. Is the manuscript presented in an intelligible fashion and written in standard English?

Reviewer #1: Yes

Reviewer #2: (No Response)

6. Review Comments to the Author

Reviewer #1: The changes recommended have been included in the revision

The article is acceptable for publication

Reviewer #2: (No Response)

7. PLOS authors have the option to publish the peer review history of their article (what does this mean?). If published, this will include your full peer review and any attached files.

Reviewer #1: No

Reviewer #2: **Yes: **Manu Chopra

---

## [Editor Report · Acceptance letter]

3 Jun 2024

PONE-D-24-04130R1 

PLOS ONE

Dear Dr. Mól, 

I'm pleased to inform you that your manuscript has been deemed suitable for publication in PLOS ONE. Congratulations! Your manuscript is now being handed over to our production team.

Kind regards, 

on behalf of

Dr. Vijay Hadda 

Academic Editor

PLOS ONE